# Influence of Housing Systems on Physical, Emotional, and Cognitive Functions with Aging in DBA/2CrSlc Mice

**DOI:** 10.3390/ani10040746

**Published:** 2020-04-24

**Authors:** Chikako Shimizu, Yoshihisa Wakita, Youichi Tsuchiya, Toshitaka Nabeshima

**Affiliations:** 1Frontier Laboratories for Value Creation, SAPPORO HOLDINGS LTD., 10 Okatome, Yaizu, Shizuoka 425-0013, Japan; Yoshihisa.Wakita@sapporoholdings.co.jp (Y.W.); Yoichi.Tsuchiya@sapporoholdings.co.jp (Y.T.); 2Advanced Diagnostic System Research Laboratory, Fujita Health University,1–98 Dengakugakubo, Kutsukake-cho, Toyoake, Aichi 470-1192, Japan; tnabeshi@ccalumni.meijo-u.ac.jp; 3NPO Japanese Drug Organization of Appropriate Use and Research, 3-1509 Omoteyama, Tenpaku-ku, Nagoya, Aichi 468-0069, Japan

**Keywords:** housing system, anxiety-like behavior, body weight, physical performance, rectal core temperature

## Abstract

**Simple Summary:**

Many scientists carefully monitor the experimental protocols, mouse strain , use of group-housing, and atmospheric enrichment in a housing-cage, but not commercially available housing-systems. The environmental conditions of mice as well as humans affects their emotional behaviors or physical activities. However, limited information is available regarding the influence of housing systems on experimental data. We used two types of housing system in the same laboratory. The difference in the structure of the two cages (chamber vs. individually ventilated cages: IVC) was whether the mouse could dangle or not. The dangling increases the amounts and quality of physical activities. Using the two-different housing systems, we investigated whether differences in physical, emotional, and cognitive functions can be observed in mice with aging. The IVC group demonstrated significantly less food intake, higher body weight, lower rectal core temperature, less muscle and balancing powers with aging, and fewer anxiety-like behaviors than the chamber group. Based on this experiment, the daily physical activities derived from housing systems significantly affected the results of body weight, body temperature, as well as their behaviors. Scientists should pay attention to the structure of housing systems and experimental parameters, particularly when changing the housing systems.

**Abstract:**

Environmental conditions, including enrichment and stress, affect animal behaviors, but limited information is available regarding the differences in animal functions between the chamber (ventilated system) vs. IVC (individually ventilated cages) housing systems. Therefore, the effects of different housing systems were examined on physical, emotional, and cognitive functions and the intestinal flora with aging. DBA/2CrSlc mice were divided into chamber and IVC groups. Differences in the structure of the two cages considered whether the mouse could dangle or not. Physical, emotional, and cognitive functions were examined using the open field, black and white box, object recognition, horizontal bar, wire hanging, balancing, footprint, and locomotor tests. The IVC group demonstrated significantly less food intake, higher body weight (by approximately 5 g), lower rectal core temperature, less muscle and balancing powers with aging, and fewer anxiety-like behaviors than the chamber group. No differences were observed in the cognitive function and intestinal microbiota between the groups. The housing environment affected the rodent basal temperature and body weight as well as the physical and emotional functions. Scientists should be attentive to the type of cages used in the housing system for an experiment, especially when comparing the results with animals reared in different systems.

## 1. Introduction

In mice, several types of housing systems have been used in different laboratories with housing conditions affecting their behaviors. For example, maternal separation, adolescent stress [1,2], and social defeat stress [3,4] lead to depression-like behaviors. In contrast, environmental enrichment demonstrates positive effects on their mental and physical well-being [5,6,7], learning and memory [8,9], and lifespan [10]. Besides proactive changes in housing conditions, the number of mice per cage [11,12] and depth of cage bedding [13] also affects rodent behavior. Recently, “individually ventilated cages” (IVC) have been used in many laboratories owing to several benefits such as the higher density housing in limited space and better protection from biohazards and allergens than available alternatives. In the IVC type, mice are isolated from the stressors of other mice, such as smell and sounds. Conversely, in the open rack and chamber systems (ventilated type), the mice can see, smell, and hear other mice. As rodents are sensitive to experimental circumstances, there is ample information regarding behavioral differences during long term breeding in Chamber and IVC systems. Among IVC systems, the differences in the housing system affect maternal performance, pup development [14], the reproductive performance of genetically engineered mice [15], and hygiene conditions of mouse colonies [16]. Among the different types of cages, Tsai et al. have compared an IVC rack with enrichment to a ventilated cabinet and normal open rack for the breeding performance [17]. Furthermore, Feistenauer et al. have reported the influence of five types of cages on allergen exposure [18]. Scientists carefully monitor the experimental protocols, the strain of mice, group-housing or individual-housing, and environmental enrichment. However, there is little or no information regarding the effects of housing systems on the accuracy of experiments. Housing systems are expensive. Therefore, once introduced into a laboratory, they are continuously used over a long time. When old housing systems are replaced by a new one, it is difficult to know how the differences between the old and new housing systems affect the quality of data.

The DBA/2 strain rapidly develops a robust sensitization to ethanol compared to other strains. Previously, we have examined the rewarding degrees of moderate drinking using the conditioned place preference (CPP) test as well as the sensitization of locomotor activities [19,20], and then evaluated behaviors in DBA/2 mice with aging. The CPP test is one of the most commonly used tests to evaluate the degree of reward and addiction to drugs [21,22,23] and foods [24,25] such as oil and sugars. The CPP equipment consists of a black and a white box, placed inside light- and sound-attenuating chambers equipped with a fan for reducing sound and smell. In well-established CPP protocols, the pre-test value is calculated based on “how long each mouse spends in the white or the black box”. As the value is directly used as the rewarding degree, the value largely affects the outcome of the CPP test.

Furthermore, we have focused on the effects of foods on healthy aging, one of the most important societal challenges due to the rapid progression of the aging society. Cognitive defects like Alzheimer’s dementia [26] and locomotive syndrome [27] are serious problems in the super-graying society. However, many aging studies have considered some limited physiological changes at only a few time points within a certain period and presented limited data. DBA/2 mice were frequently used for reward-based tests in our previous experiments [19,20]. Therefore, the optimal conditions of various behavioral tests including physical, emotional, and cognitive tests with aging were investigated in DBA/2 mice. Moreover, we revealed enrichment methods using suitable conditions for the welfare of the DBA/2 strain. To evaluate healthy aging, we investigated physical behavioral tests, such as locomotor activities, and the object location memory and object recognition memory [28,29]. Both memory functions are non-invasive under conditions similar to those utilized for human cognitive assessment. Based on experiments in DBA/2 mice, we then applied the methods in senescence-accelerated mouse (SAM) prone-strains 8 and 1 established by Takeda et al. [30,31], and the healthy aging effects of moderate drinking [32], lemon polyphenols [33], and barley [34] on changes in phenotypes have been evaluated. However, in previous experiments, we have used only DBA/2 and SAM mice reared in “a chamber type housing system.” Substantial data on whether changes in the physical, emotional, and cognitive functions with aging are similar between the two-different housing systems (chamber vs. IVC) are lacking.

The difference in the housing structure of the two cages was whether the mouse could dangle or not, suggesting differences in three-dimensional physical activities like jumping, and the frequency of clutching behaviors with the fore- and hind-feet for dangling. Therefore, we investigated whether the difference of the housing system (chamber vs. IVC) affected physical, emotional, and cognitive functions with aging. In this experiment, we used DBA/2CrSlc mice as we are familiar with daily behaviors during their lives. Furthermore, we investigated whether this difference affected other parameters, such as food consumption, body weight, rectal core temperature, physical performances in balancing abilities, anxiety-like behaviors, and the intestinal microbiome with aging. Anxiety-like behaviors were evaluated using the open field test and the black-white box. The parameters of former were the locomotor activities and the number of fecal droppings induced by brightness-induced anxiety. The parameter of latter was measured at the same conditions of the CPP pre-test. To our knowledge, this is the first report demonstrating the effects of daily physical activities induced by the different housing systems, the chamber type and IVC, on the physical, emotional, and cognitive functions of mice with aging.

## 2. Materials and Methods 

### 2.1. Ethics Statement

All experiments were approved by the Institutional Animal Care and Use Committee of SAPPORO HOLDINGS LTD. (permit numbers 2017-001, 2018-002) following the Guidelines for the Proper Conduct of Animal Experiments of the Science Council of Japan.

### 2.2. Housing Systems and Cages (IVC type vs. Chamber Type)

Mice were reared in two different housing systems and cages: chamber type, Ebac L and Clean S (CREA Japan, Inc, Tokyo, Japan); the IVC type, Innorack and Innocage OMV1 (Innovive, San Diego, CA, USA). The difference in the structure of the two cages was the lids that enabled the mouse to dangle (chamber type), or not (IVC type) (Figure 1).

### 2.3. Animals and Housing Conditions

Six-week-old DBA/2 CrSlc male mice (n = 36, Japan SLC, Inc., Hamamatsu, Japan) were divided into two groups: the chamber group (n = 18) and the IVC group (n = 18) and were reared in their cages. The presence of female animals in the same room could affect behaviors of male mice. Therefore, only male DBA/2 mice were used in this experiment. The bottom of each cage was covered with sliced paper bedding (Palmas *μ*^®^ Material Research Center, Tokyo, Japan). The animals in both groups were group-housed (six per cage) with access to water and standard chow (CRF-1, Charles River Laboratories, Yokohama, Japan) ad libitum. At 14-weeks of age, the animals started to fight, and aggressive or tail-injured mice were isolated to avoid physical injury. At 38-weeks, the fighting ceased in all cages. The number of the final housing was as follows: 2 cages of group-housing (n = 3, n = 3) and 12 isolated cages in the chamber group and 2 cages of group-housing (n = 4, n = 3) and 11 isolated cages in the IVC group. There are no difference in the spontaneous behaviors such as eating foods and approaching behavior to the enrichment between the single and group housed animals [12], and the number of groups and single housing cages were almost the same in both groups. From 19 weeks of age, the environment was enriched by placing pieces of dried and bleached dishcloth gourd into each cage over 96 w. The animal facility was maintained at 23 ± 1 °C with 55% humidity and a 12-h/12-h light/dark cycle (lights on from 7:30 a.m. to 7:30 p.m.).

Food consumption, body weight, rectal core temperature, physical, emotional, and cognitive functions in the mice were observed (Appendix A). Moreover, champing behaviors to slice papers of bedding at 6 weeks of age, and dishcloth gourd for the enrichment in cages at 20 weeks of age were observed (Appendix A).

### 2.4. Food Consumption and Body Weight 

Food consumption per cage and body weight of each mouse were recorded once or twice a week. The food consumption was measured weekly per cage. The amount of food consumed was expressed per mouse per day, which was calculated by dividing the total consumption by the number of mice per cage.

### 2.5. Measurement of Rectal Core Temperature

Body temperature mostly depends on the circadian phase and physical activity [35,36]. The mouse rectal core temperature was measured at the ages of 43, 58, 80, and 94 w from 1:00 p.m. to 3:00 p.m., the period of lowest rectal temperature [37], using a microprobe thermometer (BAT-12, Physitemp Instruments, LLC, NJ, USA). The numbers of alive mice (the chamber group, the IVC group) were 34 (n = 16, n = 18) at 43 w, 32 (n = 16, n = 16) at 58 w, 32 (n = 16, n = 16) at 80 w, and 26 (n = 14, n = 11) at 94 weeks of age, respectively.

### 2.6. Changes in Physical Performances with Aging (Tests 1–5)

To assess the physical abilities, balance, climbing potential, prehensile strength, gait abnormality, and motor coordination were observed in mice using five physical performance tests: balancing abilities on an acryl rod (Test 1) [34,38], horizontal bar (Test 2) [39], wire hanging tests (Test 3) [34,40], footprint test (Test 4) [34,41], and locomotor activities (Test 5) [33,34], respectively.

The balancing abilities on an acryl rod (Test 1), the horizontal bar test (Test 2), and wire hanging test (Test 3) were conducted at the ages of 12, 17, 24, 31, 41, 53, 60, 67, 78, 86, 92, and 96 w. The footprint test (Test 4) was performed at the ages of 13, 29, 50, 64, 77, and 93 w. The locomotor activities (Test 5) were measured at the ages of 11, 15, 33, 47, 64, 75, 82, 90, and 96 w. Tests 1, 3, and 4 were conducted as in our previous report [34].

The numbers of alive mice (the chamber group, the IVC group) of Tests 1–3 were 36 (n = 18, n = 18) at 12 w, 36 (n = 18, n = 18) at 17 w, 36 (n = 18, n = 18) at 24 w, 35 (n = 17, n = 18) at 31 w, 34 (n = 16, n = 18) at 41 w, 33 (n = 16, n = 17) at 53 w, 32 (n = 16, n = 16) at 60 w, 32 (n = 16, n = 16) at 67 w, 32 (n = 16, n = 16) at 78 w, 31 (n = 16, n = 15) at 86 w, 27 (n = 15, n = 12) at 92 w, and 25 (n = 14, n = 11) at 96 weeks of age, respectively.

#### 2.6.1. Balancing Ability on an Acryl Rod (Test 1)

The balancing abilities on an acryl rod (Test 1) [38] were modified as follows. A square 5 mm, 320 mm long-acryl rod was bridged on the top edges of a box (300 × 300 × 350 mm (D × W × H)), the bottom of which was covered with approximately 3-cm thick Palmas *μ*^®^. The mouse was perpendicularly placed on a rod and allowed to balance. The staying ability of the mouse was assessed on the rod until 180 s. This trial was repeated three times, and the maximum time recorded was used [34].

#### 2.6.2. Horizontal Bar Test (Test 2)

The horizontal bar test (Test 2) [39] was modified as follows. A 300 mm metal wire (diameter of 2 mm) was bridged on the top edges of a box (300 × 300 × 350 mm (D × W × H)), the bottom of which was covered with approximately 3 cm thick Palmas *μ*^®^. We allowed the mouse to grip the center of the wire, supporting the mouse’s tail with a hand, and then let go of the tail. The physical ability was scored using five grades as follows:
Score 5, crossing on the bar with two forefeet and two hindfeet; Score 4, not crossing the bar, but clinging with two forefeet and two hindfeet; Score 3, clinging with two forefeet and one hindfoot; Score 2, hanging with two forefeet for more than 10 s and falling off the bar; Score 1, hanging with two forefeet for under 10 s and falling off the bar.


#### 2.6.3. Wire Hanging Test (Test 3)

The wire hanging test (Test 3) [40] was modified as follows. A mouse was placed on a metal salamander with a 1-cm square mesh 50 cm above a cushioned surface covered with approximately 3-cm thick Palmas *μ*^®^. The salamander was inverted and the mouse was attached upside down. The time to fall was measured until 60 s. This trial was repeated three times and used the maximum time [34].

#### 2.6.4. Footprint Test (Test 4)

The foot print test can detect gait abnormalities [41]. The footprint test (Test 4) was modified as follows. To prepare apparatus for the foot print test, we inserted a reed-shaped white paper into a cardboard cylinder (10 cm wide, 76 cm long). The mice were allowed to walk along a runway of reed-shaped white paper inside a cardboard cylinder (10 cm wide, 76 cm long). All mice were given a few training runs for habituation and one test run after painting the four feet using children’s paints (red and blue) (Pentel Co., Ltd., Tokyo, Japan). We measured the “Stride, Sway, and Stance” lengths of both the forefeet and hindfeet of the mouse using the footprints [34].

The numbers of alive mice (the Chamber group, the IVC group) were 36 (n = 18, n = 18) at 13 w, 35 (n = 17, n = 18) at 29 w, 33 (n = 16, n = 17) at 50 w, 32 (n = 16, n = 16) at 64 w, 32 (n = 16, n = 16) at 77 w, and 26 (n = 15, n = 11) at 93 weeks of age, respectively.

#### 2.6.5. Locomotor Activities (Test 5)

The locomotor activities (Test 5) were measured for 10 min, in boxes (300 × 300 × 350 mm (D × W × H); Brain Science Idea, Inc., Osaka, Japan) in which the bottom was covered with Palmas *μ*^®^ and quantified using the ANY-maze Video Tracking System (Stoelting Co., Wood Dale, IL, USA). The boxes were illuminated at 25 lux as described in our previous reports [33,34]. The time spent in the center area (center time) of the box was measured for 10 min. The numbers of alive mice (the chamber group, the IVC group) were 36 (n = 18, n = 18) at 11 w, 36 (n = 18, n = 18) at 15 w, 35 (n = 17, n = 18) at 33 w, 33 (n = 16, n = 17) at 47 w, 32 (n = 16, n = 16) at 64 w, 32 (n = 16, n = 16) at 75 w, 31 (n = 16, n = 15) at 82 w, 29 (n = 16, n = 13) at 90 w, and 25 (n = 14, n = 11) at 96 weeks old age, respectively.

### 2.7. Changes in Anxiety-like Behaviors with Aging (Test 6 and Test 7)

#### 2.7.1. Black and White Box Test (Test 6)

Limited information is available regarding the relationship between the housing system and the anxiety behaviors in mice. The black and white test was performed to assess the exploratory behavior relevant to anxiety [42]. In this test, we utilized the experimental condition identical to the pre-test condition of CPP in our previous reports [19]. Black and white box test (Test 6): The illuminating conditions in black and white boxes (300 × 150 × 150 mm; Brain Science Idea Inc., Osaka, Japan) were off-scale low and 125 lux, respectively, as the black chamber was covered with blacktop. Movements were scored by both video tracking and a stopwatch to measure the time spent in the white area for 900 s. The number of times the mouse entered the black and white compartments of the box was measured for 900 s. The numbers of alive mice (the chamber group, the IVC group) at the ages were 36 (n = 18, n = 18) at 8 w, 36 (n = 18, n = 18) at 24 w, and 33 (n = 16, n = 17) at 51 weeks of age, respectively.

#### 2.7.2. Open Field Test (Test 7)

The open-field test is another evaluation for the anxiety behavior [43]. Open field test: The anxiety-like behaviors were evaluated using an open field test (Test 7). Test 7 was performed in the same boxes as Test 5, except using a smooth floor by removing Palmas *μ*^®^ from the bottom and at 200 lux (brightness-induced anxiety). The locomotor activities, center time (same as in Test 5), and the number of fecal droppings were measured.

The numbers of alive mice (the chamber group, the IVC group) were 32 (n = 16, n = 16) at 58 w, 31 (n = 16, n = 15) at 87 w, and 26 (n = 15, n = 11) at 93 weeks of age, respectively.

### 2.8. Changes in Spatial Recognition (Test 8, Short-Term Location Memory) and Object Recognition (Test 9, Long-Term Object Memory) with Aging

To assess the object location memory and object recognition memory, the object location test (OLT) and the object recognition test (ORT) were conducted, respectively. The cognitive functions of the OLT (Test 8) were observed at 16 and 33 weeks of age and the novel ORT (Test 9) at of 34, 48, 65, 76, and 91 weeks of age, as previously reported [32,33,34,44].

#### 2.8.1. Paradigm of ORT and OLT (Test 8 and Test 9)

ORT and OLT consisted of the following 3 phases: habituation, training, and test (Figure 2).

Habituation: Each mouse was free to explore the ORT box without objects for 10 min, once a day for 3 consecutive days during the ORT and OLT habituation phase.

Training: In this phase, two of the identical objects (A) were fixed to the floor. In the ORT and OLT training phases, mice were placed in the box and allowed to freely access the two objects for 10 min.

Test: In the test phase, the right object was placed at a different position (A’) for OLT or replaced by a novel object (B) for ORT. The interval between the training and test phase was 24 h for the assessment of long-term memory during the ORT or 2 h for short-term memory during the OLT.

For ORT objects, white golf balls (43 mm diameter) and a white film case (29 mm diameter × 50 mm height) were used during the training and the test phases, respectively. For OLT objects, a wooden apple block (31 mm width, 50 mm height of the main body and 15 mm height of stem end) at 16 weeks of age and a green cylindrical wooden block (44 mm diameter and 44 mm tall placed horizontally) at 33 weeks were used.

#### 2.8.2. Data analysis of ORT and OLT (Test 8 and Test 9)

In the training phase, the recognition index of right and left objects during the training phase for each mouse (Figure 2) was expressed as the ratio of the amount of time spent exploring object left A (Time left A × 100)/(Time left A + Time right A) and the amount of time spent exploring object right A (Time right A × 100)/(Time left A + Time right A) for both ORT and OLT.

During the test phase of OLT, the recognition index for each mouse was expressed as the ratio of the amount of time spent exploring familiar location of object A (Time A × 100)/(Time A + Time A’) and the amount of time spent exploring novel location of object A’ (Time A’ × 100)/(Time A + Time A’).

During the test phase of ORT, the recognition index for each mouse was expressed as the ratio or amount of time spent exploring familiar object A (Time A × 100)/(Time A + Time B) and the amount of time spent exploring novel object B (Time B × 100)/(Time A + Time B).

Differences between recognition indexes of the left and right (or novel location) objects were assessed using the unpaired t-test for OLT (ORT) in each phase [32].

### 2.9. Changes in Intestinal Microbiome

Reportedly, the intestinal microbiome changes due to several factors including age [45], foods [32,33,34], and the forced exercise [46].

Fresh feces (approximately 100 mg) were collected, stored at −30 °C, and used to analyze the intestinal microbiome. Bacterial DNA was isolated with some modifications. Briefly, the bacterial suspension was treated with lysis buffer at 70 °C for 10 min in a water bath and vortexed vigorously for 40 s using a FastPrep 24 Instrument (MP Biomedicals, Santa Ana, CA, USA) at a speed of 6.0 m/s [47]. The subsequent procedure was performed according to our previous study [34].

UniFrac is a distance metric used for comparing biological communities and visually expresses the composition of bacterial species at a specific site [48]. UniFrac distance analysis was performed, and the proportion of the intestinal microbiome at the phylum and genus levels was determined by RDP (Ribosomal Database Project) classifier using the Greengenes database (gg_13_8_otus/taxonomy/97_otu_taxonomy). Changes in the intestinal microbiome were investigated using UniFrac analysis at 17, 20, 29, 38, 47, and 59 weeks of age.

The numbers of alive mice (the chamber group, the IVC group) were 36 (n = 18, n = 18) at 6 w, 36 (n = 18, n = 18) at 17 w, 36 (n = 18, n = 18) at 20 w, 35 (n = 17, n = 18) at 29 w, 34 (n = 16, n = 18) at 38 w, 34 (n = 16, n = 18) at 47 w, and 32 (n = 16, n = 16) at 59 weeks of age, respectively.

### 2.10. Degree of Visceroptosis

The degree of visceroptosis was scored by examining the bulge of the mouse-rump at 94 weeks of age and counting the number of mice according to 3 degrees: (−), no bulge; (+), a little bulge; (++), a large bulge of the rump (Appendix A).

### 2.11. Statistical Analyses 

All statistical analyses were performed using the SPSS software 10.0.7J for Windows (SPSS, Inc., Chicago, IL, USA). Data in the text and figures are presented as the mean ± standard error (SE) of the mean. For the acryl rod, footprint, horizontal bar, foot print and locomotor activity tests, between-group comparisons were performed using both the two-way repeated measures ANOVA using only alive mice until the last measurement, and the two-way analysis of variance (ANOVA) followed by Bonferroni’s *post-hoc* test for multiple comparisons using all mice for the healthy aging effects if the interactions were significantly different. Between-group comparisons of food consumption, body weight, rectal core temperature, open field test, ORT, and OLT were performed using a t-test. The age comparison of the black and white test, and the open field test was performed using one-way ANOVA followed by Tukey’s *post-hoc* test. The within-group comparison of body weight in the IVC group was performed by using a paired t-test. For all analyses, a *p*-value < 0.05 was considered statistically significant.

## 3. Results

### 3.1. Food Consumption and Body Weight with Aging

The food consumption in the chamber group was higher than that in the IVC group during the lifetime (*p* < 0.05, at 11–15, 26, 38, 40–47, 60–62, 64, 66, 68, 70, 73, 79–82, 85–91, and 94–96 weeks of age) (Figure 3A), whereas the body weight in the chamber group was significantly lower than that in the IVC group from seven to 96 weeks of age (*p* < 0.05), except at 92 w. The maximum difference in body weight between both groups was 5.6 g (approximately 20% heavier, *p* < 0.001) at 40 and 64 weeks of age (Figure 3B). In contrast, averages of body weight in both groups, especially in the IVC group, significantly decreased from 20 to 30 weeks of age (*t* (17) = 2.27, *p* = 0.037, by paired t-test) after the mice started fighting, and recovered when the fighting mice were completely isolated at 38 w. After the isolation, the body weights gradually increased in both groups and mice were significantly heavier in the IVC group than in the chamber group (*p* < 0.05).

### 3.2. Changes in Rectal Core Temperature with Aging

From 43 to 94 weeks of age, the rectal core temperature in the chamber group was significantly higher than that in the IVC group (43 w; *t* (24) = −3.44, *p* = 0.002, 58 w, *t*(30) = −3.37, *p* = 0.002, 80 w; *t*(30) = 2.63, *p* = 0.013, 94 w; *t* (23) = −2.96, *p* = 0.007). The maximum difference in the rectal core temperature was 1 °C. Moreover, the temperature in both groups decreased with aging (Figure 4). From 58 to 94 weeks of age, the temperature decreased more than 1 °C with aging in both the IVC (*t* (27) = 3.57, *p* = 0.001) and chamber groups (*t* (28) = 3.35, *p* = 0.002). Conversely, no significant differences were observed in the rectal core temperature between the group-housing and the isolated mice from 43 to 94 weeks old (data not shown, 43 w; *t* (32) = −0.87, *p* > 0.05 , 58 w, *t* (30) = −0.76, *p* > 0.05, 80 w; *t* (30) = 0.35, *p* > 0.05, 94 w; *t* (23) = −0.78, *p* > 0.05).

### 3.3. Changes in Physical Performances in chamber and IVC groups with Aging (Tests 1–5)

#### 3.3.1. Balancing Ability on an Acryl Rod (Test 1)

The ability significantly decreased with aging (Figure 5A, *p* < 0.001) by the repeated measures two-way ANOVA to assess the effects of groups and ages (from the ages of 12 to 96 w) on the balancing ability on an acryl rod. There were significant differences between both groups by the two-way ANOVA (*p* < 0.001), but not by the repeated two way ANOVA (*p* > 0.05). There were no significant interactions between group and age (*p* > 0.05, Table 1).

#### 3.3.2. Horizontal Bar Test (Test 2)

Based on the repeated measures two-way ANOVA to assess the effects of groups and ages on the climbing potential using the horizontal bar test, the climbing potential significantly decreased with aging (Figure 5B, *p* < 0.001), and the potential in the chamber tended to be higher than that in the IVC group (*p* = 0.054). There were significant differences between both groups by the two-way ANOVA (*p* < 0.001). No significant interactions were observed between group and age (*p* > 0.05, Table 1).

#### 3.3.3. Wire Hanging Test (Test 3)

By the repeated measures two-way ANOVA and to assess the effects of groups and ages (from the ages of 12 to 96 w) on the prehensile strength using the wire hanging test, the prehensile strength significantly decreased with aging (Figure 5C, *p* < 0.001), and significantly differed between the chamber and the IVC groups. The prehensile strength in the chamber group significantly delayed aging (Figure 5C, *p* < 0.001). A significant interaction was observed between group and age (*p* < 0.05, Table 1).

For the between-group comparison, the two-way ANOVA followed the Bonferroni’s *post-hoc* test was performed. The time to fall was significantly longer in the chamber group than in the IVC group from 12 to 96 weeks of age (*p* < 0.05), except at 24 w (*p* = 0.117).

#### 3.3.4. Footprint Test (Test 4)

The results of statistical analysis for Test 4 were shown in Table 2. Based on the repeated measures two-way ANOVA to assess the effects of groups and ages (from the ages of 12 to 93 w) on the gait abnormality using the footprint test, the stride and sway of the forefeet (Figure 6A,B) and hind-feet (Figure 6D,E) significantly decreased with aging (*p* < 0.05). The stance of the fore- and hindfeet (Figure 6C,F) significantly changed with aging (*p* < 0.05). Significant differences between groups were observed in the stride of fore- and hind-feet (Figure 6A,D), and the sway of hindfoot (Figure 6E). There were never significant interactions between group and age in the stride, sway, and stance of fore- and hindfeet (*p* > 0.05, Table 2).

#### 3.3.5. Locomotor Activities (Test 5)

Based on the the repeated measures two-way ANOVA to assess the effects of groups and ages (from 11 to 96 weeks of age) on motor coordination using locomoter activities, the locomoter activities were changed with aging (Figure 7A, *p* < 0.001), and the center time significantly increased with aging (Figure 7B, *p* < 0.001). There were no significant differences between both groups in locomoter activity and the center time (*p* > 0.05, Table 3), and no significant interactions between age and group in locomoter activity (*p* > 0.05, Table 3), but not in the center time (*p* < 0.01, Table 3).

### 3.4. Changes in Anxiety-like Behaviors with Aging (Test 6 and Test 7)

#### 3.4.1. Black and White Box Test (Test 6)

Although mice were tested after a three-day habituation, the time spent in the white area was significantly higher in the IVC group than in the chamber group at eight weeks of age (*t* (34) = 2.31, *p* = 0.027, Figure 8A). The number of crossings times mice crossings between the the black and white areas appeared to be higher (*t* (34) = 2.01, *p* = 0.052, Figure 8B) in the IVC group than in the chamber group, and were significantly higher in the IVC group than in the chamber group at 24 weeks of age (*t* (34) = 2.70, *p* = 0.011, Figure 8B). No significant difference observed between groups at 51 w (*t* (31) = 0.25, *p* > 0.05).

In one-way ANOVA followed by Tukey’s *post hoc* test, no significant change was observed with aging in both time spent in the white area (chamber group; *p* > 0.05, IVC group; *p* > 0.05) and the number of crossings (chamber group; *p* > 0.05, IVC group; *p* > 0.05) (Figure 8A,B).

#### 3.4.2. Open Field Test (Test 7)

Under Test 7 conditions, the anxiety-like behaviors were induced by an unfamiliar environment when compared to the home cages. In the IVC group, the locomotor activities were significantly higher (*t* (30) = 2.34, *p* = 0.026, Figure 9A), but the number of fecal droppings was significantly lower, than those in the chamber group at 58 weeks of age (*t* (23) = −2.16, *p* = 0.041, Figure 9C). No significant differences were observed in the locomotor activities and the number of fecal droppings between both groups at 87 and 93 weeks of age. From 58 to 93 weeks of age, the center time in the IVC group did not differ from that observed in the chamber group (Figure 9B).

In the within-group comparisons, the locomotor activities in the chamber group significantly increased from 58 to 87 w (*p* = 0.021) and those in the IVC group significantly decreased from 58 to 93 w (*p* < 0.001) in one-way ANOVA followed by Tukey’s post hoc test (Figure 9A). The center time in the chamber group decreased significantly from 87 to 93 weeks old (*p* = 0.012, Figure 9B). The number of feces significantly increased from 58 to 93 weeks of age (*p* = 0.002) in the IVC group, whereas it increased significantly from 87 to 93 weeks (*p* = 0.005) in the chamber group (Figure 9C).

### 3.5. Changes in Spatial Recognition (Test 8, Short-Term Location Memory) and Object Recognition (Test 9, Long-Term Object Memory) with Aging

The OLT was conducted to examine spatial recognition in mice at 16 and 33 weeks of age. After three days of habituation, in the training phase, the total time approaching the two identical objects was significantly higher in the chamber group than that in the IVC group (21.7 ± 1.6 s in the chamber group vs. 16.7 ± 1.2 s in the IVC group, *t* (70) = −2.43, *p* = 0.018) at 16 weeks of age (data not shown). In the test phase, significant differences in the recognition index between familiar and novel locations were observed in the chamber and IVC groups at 16 w (*t* (34) = −3.27, *p* = 0.002, *t* (34) = −2.98, *p* = 0.005, respectively), but not at 33 weeks of age (*t* (32) = 0.62, *p* > 0.05, *t* (34) = −1.25, *p* > 0.05, respectively) (Figure 10A,B). In the training phase, the recognition indexes for the two identical objects should be 50%:50% (right and left objects). The recognition indexes of ORT in the chamber group were approximately 50%:50% (i.e., no significant difference between two objects in the training phase) from 33 to 91 weeks of age (Figure 10C). However, in the IVC group, the recognition indexes were significantly different at 48 and 65 weeks of age (*t* (32) = 3.24, *p* = 0.002, *t* (30) = 2.33, *p* = 0.027, respectively)(Figure 10D). In the test phase, significant differences were observed in the ORT conducted from 34 to 91 weeks of age and the recognition index for the novel object (film case) was significantly higher than that for the familiar object (a golf ball) in both the chamber and IVC groups (34 w; *t* (32) = −25.1, *p* < 0.001, *t* (32) = −32.4, *p* < 0.001, 48 w; *t* (30) = −23.5, *p* < 0.001, *t* (32) = −16.9, *p* < 0.001, 65 w; *t* (30) = −17.7, *p* < 0.001, *t* (30) = −10.7, *p* < 0.001, 76 w; *t* (30) = −12.4, *p* < 0.001, *t* (30) = −11.9, *p* < 0.001, 91 w; *t* (30) = −12.5, *p* < 0.001, *t* (22) = −12.5, *p* < 0.001, respectively) (Figure 10C,D).

### 3.6. Changes in Intestinal Microbiome with Aging 

In mice, the overall structure of the intestinal microbiome was evaluated by unweighted UniFrac analysis using all data at six (3 d after housing in our laboratories), 17, 20, 29, 38, 47, and 59 weeks of age. Figure 11A,B reveals the same UniFrac data with the colors indicating the ages (Figure 11A) and chamber and IVC groups (Figure 11B). At first, the intestinal microbiome largely changed with aging from six to 17 w and then gradually changed with aging in the direction of the arrow (Figure 11A). However, no difference was observed in the intestinal microbe between the groups (Figure 11B).

### 3.7. Changes in Other Behaviors and Body Features with Aging 

#### 3.7.1. Champing Behavior to Slice Papers of Bedding in Cage and Dishcloth Gourd for Enrichment 

For bedding, we used sliced papers for both groups. Three days after the introduction to our laboratories, the IVC group stretched most slice papers by champing, but not the chamber group (Appendix A). The housing cages were enriched using pieces of dishcloth gourd from 19 weeks of age. Furthermore, the IVC group split the dishcloth gourds. This behavior was not observed in the chamber group (Appendix A). Thus, the champing behavior in the IVC group was more pronounced than that in the chamber group.

#### 3.7.2. Changes in the Bulge of the Rump at the Age of 94 W 

At 94 weeks of age, the mice numbers in the chamber and the IVC groups were 15 and 11, respectively. We observed the difference in visceroptosis and scored it using three degrees (−, +, and ++) at 94 w (Appendix A photos; upper: general body, lower: the bulge of the rump). No significant differences were observed in the degree of visceroptosis between the groups, but the percentage of ++ in the chamber group was higher than that in the IVC group (33% vs. 18%, Appendix A).

## 4. Discussion

Both the chamber and IVC were placed in the same room and temperature and humidity were controlled in the same manner. Both groups of DBA/2CrSlc mice were fed the same feed (CRF-1) and tap water and reared using the same sliced paper bedding in their cages. Therefore, the experimental results may be mainly induced due to the differences in the housing systems. The largest difference between both chamber and IVC systems was the structure of cage-cover, flat and smooth, or grilled, respectively (Figure 1). The differences of structures are associated with three-dimensional physical activities like jumping, and frequency of clutching behaviors with fore- and hind-feet for dangling in the chamber group, and the two-dimensional physical activities like walking, but sometimes riding on the enrichment, i.e., low frequency of muscle used in the cage. We assumed that the daily physical activities in the chamber group were more energetic than those observed in the IVC group both in quantitative and qualitative aspects.

The body weight of the IVC group was 5 g higher than that of the chamber group (Figure 3B), although the food consumption was less than that recorded in the chamber group (Figure 3A). Obesity is caused by a chronic imbalance between energy intake and energy expenditure [35]. Jung et al. have reported a significant correlation between food consumption and distance, duration, and speed on the running wheel in SWR/J mice [49]. The lower food consumption in the IVC group (Figure 3A) may be due to low physical activity as dangling was impossible. Thus, IVC mice gained body weight more easily than in the chamber group mice (Figure 3B).

Furthermore, the rectal core temperature decreased in the chamber group by approximately 1 °C with aging, but was significantly lower in the IVC group than in the chamber group (Figure 4). The rectal core temperature is affected by not only age [50], but also by stress conditions [51]. While cohort removal reportedly affects the rectal core temperature [51], no significant differences were observed between the group-housed and isolated mice (*p* > 0.05, data not shown) in our experiment. Although mice were isolated from 14 to 38 weeks of age due to aggressive behavior in group housing, the rectal core temperature was measured at 43 w, more than one month after the stop isolation of the last mouse. Therefore, mice might acclimatize to the isolated environment, and isolation may not have directly affected the stress and rectal core temperature in this experiment. Conversely, Tsuzuki et al. have reported that the exercise affects elevation of body temperature, and activation of the Akt, a key factor in insulin signaling pathway in the skeletal muscle of type 2 diabetic rats [52]. We speculate that the difference in rectal core temperature between the groups may be related to the degree of physical activity and body weight in the different housing systems. We should be attentive to the physical activities in housing systems when comparing data from animals reared in different housing systems. Additionally, IVC systems might be suitable in experiments analyzing anti-obesity effects due to the facilitation of weight gain associated with lower exercise.

The loss of muscle strength impacts physical performance with aging [53]. The changes in physical performance in aging mice were investigated to assess balance [38], climbing potential [39], and prehensile strength [40] using an acrylic rod (Test 1, Figure 5A), horizontal bar (Test 2, Figure 5B), and wire hanging (Test 3, Figure 5C). In the three tests, the mice in the chamber group were significantly better at climbing tasks than those in the IVC group. The results of the balancing abilities (Figure 5A) and horizontal bar test (Figure 5B) closely correlated as both activities require generalized performance and forefeet strength. The difference between both groups was more evident in the wire hanging test (Figure 5C) because wire-hanging behavior is similar to clutching behavior with fore- and hind-feet as observed in dangling behavior. Although the body weight of some mice decreased during fighting, the results of Test 1–4, and Test 7 were not abnormal during the period of fighting (Figure 5A–C, Figure 6A–F). Therefore, the fighting might not affect the results of these tests.

Based on these results, the type of exercise may contribute to delay the aging of physical performances in this experiment. Moreover, housing systems, especially the structure of cages, should be considered if the aim of the experiment is to analyze physical performance and muscular strength of the fore- and hind-feet. Additionally, the bulging degree of the rump in the chamber group was greater than that in the IVC group at 94 weeks of age (Appendix A). Visceroptosis is a kind of sinking of internal organs below their natural position. The bulge of the rump may be reflected by visceroptosis due to the frequency of standing on two legs, just before dangling, which might increase and lead to visceroptosis. Thus, mice in the chamber system may suffer visceroptosis with aging like humans. Moreover, the structure of housing systems should be carefully considered as the aging body shape could be affected by daily dangling.

Regarding the footprint test (Test 4), the strides and sways, and stances for both fore- and hind-feet were significantly altered with aging (Figure 6A–F). The stride and sway distances were decreased and became bandy-legged (the increase of stance) with aging. The age-related gait reduction in DBA/2 was similar to that observed in humans [54]. Moreover, a significant difference was observed between the groups in strides and sways (Figure 6A,D,E), but not in the stances (Figure 6C,F) of footprints. The difference was observed until 20 weeks old and then disappeared. Hunter et al. have reported that well-trained runners naturally select stride rates and lengths that minimize oxygen uptake at distance running speeds [55]. In this experiment, the optimal stride length might be adapted for oxygen uptake in mice. The daily physical activities seemed to be affected in the younger mice and were maintained until middle age. The type of cage systems used should be carefully considered if the foot-print test needs to be performed using young transgenic mice [56].

In contrast, no significant difference was observed between both groups in locomotor activities under 25 lux (Test 5, Figure 7A). The housing systems did not affect the two-dimensional movement and locomotor activities (Figure 7A), unlike three-dimensional movements (Figure 5A–C). The DBA/2 mice did not demonstrate walking disorders until 96 weeks of age (not extremely old), unlike the SAMP8 mice with osteoarthritis and histological joint degeneration with aging [57]. Mice in the IVC group walked daily in the two-dimensional movement and sometimes climbed on the bleached dishcloth gourd for enrichment. If mice in the IVC group are not enriched, their locomotor defeat may appear faster. In other words, this enrichment might be effective to maintain spontaneous activity without dangling. Mice in the IVC group showed the champing behavior to slice bedding paper and split dishcloth gourd (Appendix A), instead of the dangling behavior in the chamber group. These behaviors might have been induced by the enriched housing conditions, which could have decreased stress and caused the difference between the groups.

There is limited information regarding the effects of commercial housing system on the results of anxiety-like behaviors. The anxiety-like behaviors were investigated using the black and white box [42,58] and open-field tests [43,59]. Even after the three-day habituation equal to the pre-test of CPP [19], the time spent in the white area was increased and the number of crossings between the black and white areas was significantly higher in the IVC group than in the chamber group, i.e., the IVC group preferred the bright (white) area than the dark black area, which indicated a low anxiety-behavior due to the different housing systems. The low anxiety like-behavior was significantly observed only at 8 w, decreased with aging, and disappeared at 51 weeks of age, i.e., the low anxiety-behavior in the IVC group appeared only in younger mice. The CPP test is normally conducted in young mice [21,22,23,24,25]. Thus, we should pay attention to the housing systems in the CPP test. The levels of preference in the CPP test could be influenced by the housing cage system because the time spent in the white area (Figure 8A) was used as the pre-test values of CPP. Therefore, the brightness condition in the CPP test should be adjusted such that the initial preference for black and white areas in mice is identical. Therefore, the optimal conditions of CPP may differ if the housing systems are changed in a laboratory.

Regarding the open field test (Test 7), another test assessing anxiety like-behavior, under the unfamiliar and novel conditions compared to the home cage, and without the habituation process to the test box in the open-field test, unlike Test 6, the locomotor activities were significantly higher and the number of fecal droppings was less in the IVC than in the chamber group at 58 weeks of age. Furthermore, these results supported the low anxiety like-behavior in the IVC group (Figure 9A,C). Moreover, the difference between both groups also disappeared with aging. Mice might acclimatize the experimental conditions of Tests 6 and 7 as well as the housing systems with aging. Additionally, the locomotor activities significantly decreased and the number of fecal droppings significantly increased in the IVC group from 58 to 93 weeks of age. Therefore, the changes in anxiety like-behavior with aging in the chamber group were small, but those in the IVC group were large, even when the mice expressed low anxiety like-behavior at a young age.

Even though the same locomotor equipment was used besides the brightness and the smoothness on the floor, the locomotor activities and the changes with aging were significantly affected. Pan-Vazquez et al. have reported that long term exercise enhances resilience to stress in single-housed mice, but in pair-housed mice [60], however, no significant differences in black and white box and the open field tests were observed between the group-housing- and the isolated-mice in our experiments (data not shown, *p* > 0.05). Mice in the IVC type are isolated from the stressors of smell and sounds from other mice, but those in the chamber are not. We think that the low anxiety behavior in the IVC group was induced by the difference of ability to alertness. These results indicate the importance of experimental conditions in mice. Thus, in experiments to investigate emotional behaviors, housing systems in use need to be considered.

The cognitive functions, as well as the physical performances, are essential for the evaluation of healthy aging. The ORT and OLT are commonly used and are non-invasive under conditions similar to those used for human cognitive assessment [28,29]. Cognitive functions were investigated using the spatial memory by OLT (Figure 10A,B) and the object memory by ORT (Figure 10C,D). Although no significant difference in OLT was observed between the groups at 16 and 33 weeks of age (Figure 10A,B), in the training phase, the total time spent approaching the two objects in the chamber group was significantly longer than that in the IVC group at 16 weeks of age (*p* < 0.05). The mice in the chamber systems can see, smell, and hear other mice, which led to the alertness. Therefore, the anxiety like-behavior in the chamber group might be higher than that in the IVC group in the black and white box test (Figure 8A,B). The training and test phases in ORT and OLT were conducted after the habituation phase [32,33,34]. The mice in the chamber group might know life-threatening objects during the habituation phase, and carefully approached to the object in the training and test phases due to alertness.

In the training phase, two identical objects were used. However, in the IVC group, significant differences were observed in the recognition indexes of ORT between the identical objects at 48 and 65 weeks of age (Figure 10D), but not in the chamber group (Figure 10C). In the chamber group, the ability to balance in dangling might be better for OLT and ORT, as mice equally approached the same objects in the training phase. Dond et al. have reported that the voluntary exercise improves the novel object location index [61]. Groot et al. have reported that the physical activity interventions positively influence cognitive function in patients with dementia [62]. Hyodo et al. have reported the association between aerobic fitness and cognitive function in older men [63]. In our results, no significant differences were observed in ORT between both groups with or without physical activity like dangling at 91 weeks of age. As the DBA/2 is not a dementia model like SAMP8 mice, the cognitive defeat may not progress at 91 weeks of age. However, for the cognitive evaluations, the cages with dangling in the chamber group might be more suitable than those without dangling in the IVC system, because of the longer approaching time to objects, and the better balancing ability in the training phase (Figure 10B).

In the intestinal microbiome, no remarkable difference was observed between the groups (Figure 11B) by UniFrac analysis, although the maximum difference in the rectal core temperature between both groups was approximately 1 °C. However, the intestinal microbiome in both groups was greatly altered from six to 17 weeks of age (Figure 11A). The intestinal microbiome is greatly affected by food [32,33,34], breeders [64], and stress via the gut–brain axis [65,66]. In this experiment, mice consumed different food at a breeder for six weeks after birth, after which the mice were reared using standard food (CRF-1) in our laboratory. The drastic change in the intestinal microbiome from six to 17 weeks of age could be due to the differences in foods and housing between the breeder and our laboratory. Allen et al. have reported that forced exercise including forced treadmill running alters the gut microbiome when compared with the voluntary wheel running in C57BL/6J mice [46], and no remarkable difference was observed in the intestinal microbiome between the groups. The physical activity like dangling might not be as hard as forced running to change the intestinal microbiome.

## 5. Conclusions

The difference in housing cages structures, i.e., whether the mouse could dangle or not, affected physical, behavioral, emotional, and cognitive functions. Mice in the IVC group showed less food intake, heavier body weight (by approximately 5 g), lower rectal core temperature, less muscle and balancing powers with aging, and fewer anxiety-like behaviors than those in the chamber group. The housing environment affected not only the basal temperature and body weight, but also the physical and emotional functions of mice. Scientists should adequately consider the type of cage in the housing system for an experiment, especially when compared with the results of animals reared in different systems.

## Figures and Tables

**Figure 1 animals-10-00746-f001:**
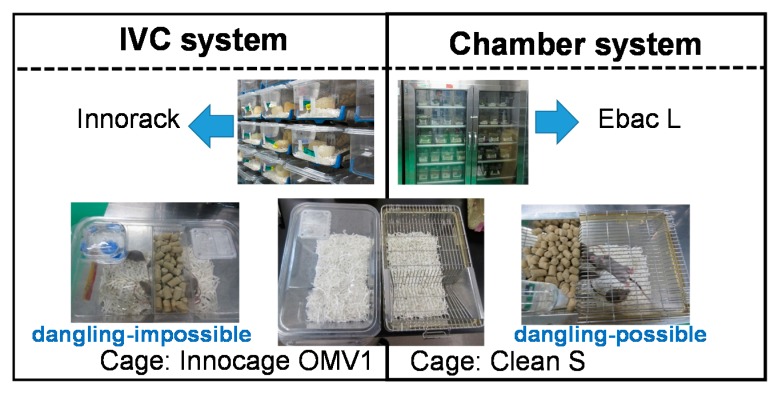
Two different types of housing cages: chamber (right) and IVC (individually ventilated cages) (left) types.

**Figure 2 animals-10-00746-f002:**
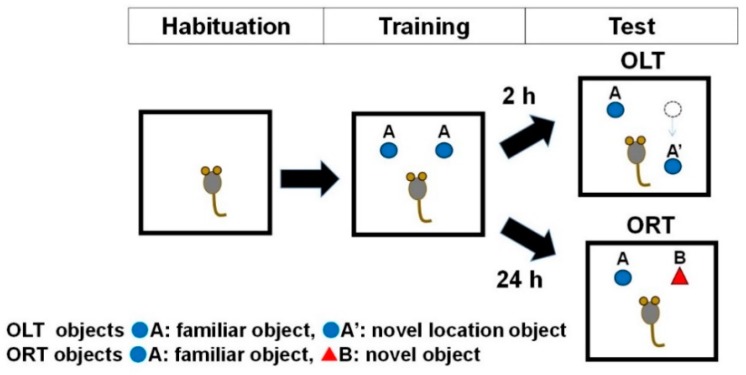
Schematic representation of the three phases (Habituation, Training, and Test Phases) for the Object Location Test (OLT) and Object Recognition Test (ORT).

**Figure 3 animals-10-00746-f003:**
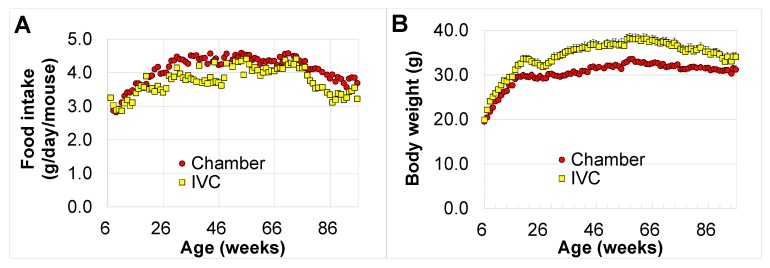
Changes in food consumption and body weight with aging in the chamber and IVC groups: (**A**) food consumption (g/mouse/day), (**B**) body weight.

**Figure 4 animals-10-00746-f004:**
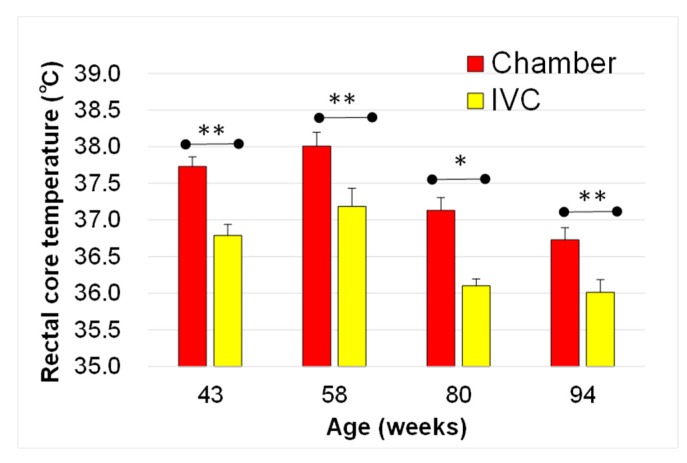
Changes in rectal core temperature with aging in the chamber and the IVC groups. ** *p* < 0.01; * *p* < 0.05 by t-test.

**Figure 5 animals-10-00746-f005:**
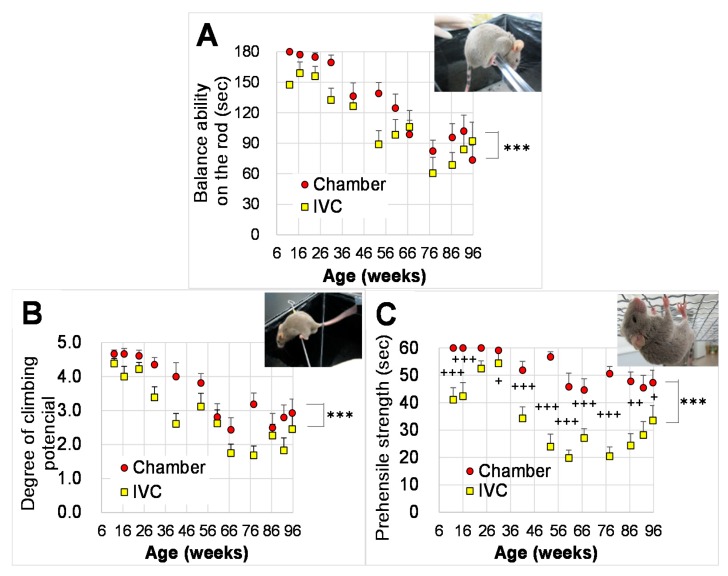
Changes in physical performance with aging in the chamber and IVC groups: physical performance in balancing abilities on (**A**) an acryl rod (Test 1), (**B**) horizontal bar (Test 2), (**C**) wire hanging (Test 3). *** *p* < 0.001 by two-way ANOVA, ††† *p* < 0.001; †† *p* < 0.01; † *p* < 0.05 by Bonferroni’s *post-hoc* test for multiple comparisons.

**Figure 6 animals-10-00746-f006:**
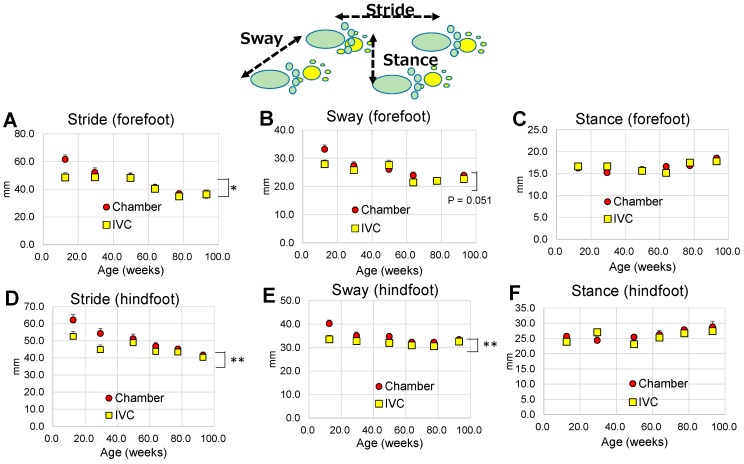
Changes in footprint with aging in the chamber and IVC groups (Test 4): Forefoot; (**A**) stride, (**B**) sway, (**C**) stance, Hindfoot; (**D**) stride, (**E**) sway, (**F**) stance. ** *p* < 0.01; * *p* < 0.05 by two-way ANOVA.

**Figure 7 animals-10-00746-f007:**
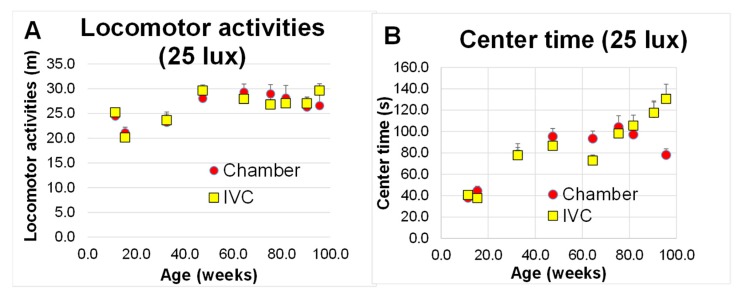
Changes in locomotor activities under 25 lux with aging in the chamber and IVC groups (Test 5): (**A**) locomotor activities; (**B**) center time. There was no significant difference between both groups using two-way ANOVA.††† *p* < 0.001 by Bonferroni’s *post-hoc* test for multiple comparisons.

**Figure 8 animals-10-00746-f008:**
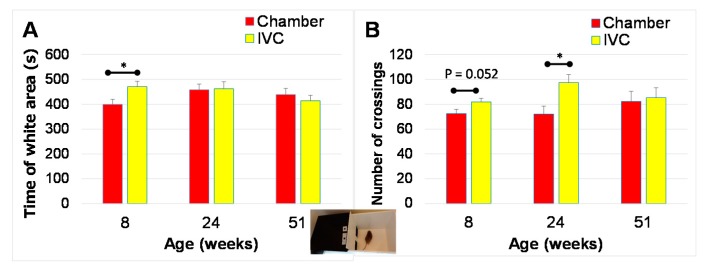
Changes in behaviors with aging in black and white box test in the chamber and IVC groups (Test 6): (**A**) Time spent in the white area for 900 s; (**B**) Number of crossings between the white and black areas for 900 s. * *p* < 0.05 by t-test.

**Figure 9 animals-10-00746-f009:**
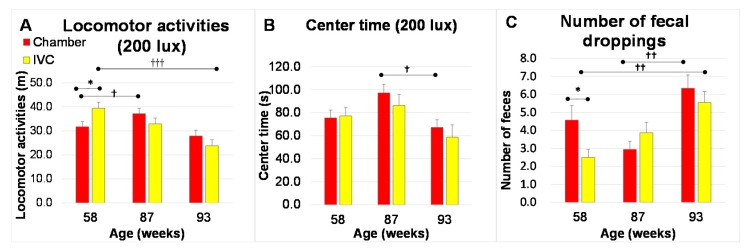
Changes in behaviors in open field test under 200 lux with aging in the chamber and IVC groups (Test 7): (**A**) Locomotor activities; (**B**) Center time; (**C**) Number of fecal droppings. Between-group comparison: * *p* < 0.05 by t-test, Within-group comparison : ††† *p* < 0.001; †† *p* < 0.01; † *p* < 0.05 by one-way ANOVA followed by Tukey’s *post hoc* test.

**Figure 10 animals-10-00746-f010:**
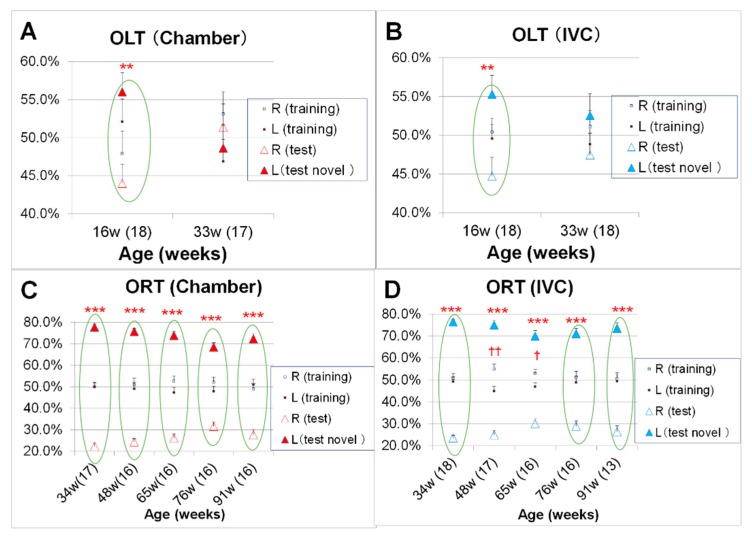
Changes in cognitive functions with aging in object location test (OLT, Test 8) and object recognition test (ORT; Test 9) in the chamber and IVC groups: ORT: (**A**) chamber group, (**B**) IVC group, ORT; (**C**) chamber group, (**D**) IVC group. Comparison of recognition index in terms of location of the familiar object (OLT) and the novel object (ORT) during the test phase using the unpaired t-test., *** *p* < 0.001; ** *p* < 0.01 by t-test. Between-group comparisons in the training phase, †† *p* < 0.01; † *p* < 0.05 by t-test.

**Figure 11 animals-10-00746-f011:**
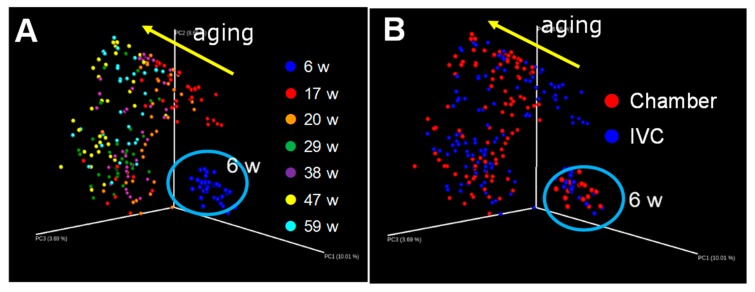
Changes in the intestinal microbiome with aging in the chamber and IVC groups: UniFrac analyses (unweighted); (**A**) ages; (**B**) groups.

**Table 1 animals-10-00746-t001:** Results of statistical analysis in Test 1–Test 3.

		Repeated Measures Two-Way ANOVA	Two-Way ANOVA
		Alive Mice at 96 Weeks of Age	All Mice
Balancing Ability on an Acryl Rod (Test 1)	age	*F* (11,253) = 20.0, *p* < 0.001	*F* (11,365) = 16.6, *p* < 0.001
	group	*F* (123) = 1.63, *p* > 0.05	*F* (1365) = 14.8, *p* < 0.001
	age × group	*F* (11,253) =1.59, *p* > 0.05	*F* (11,365) = 1.04, *p* > 0.05
Horizontal Bar Test (Test 2)	age	*F* (11,253) = 16.0, *p* < 0.001	*F* (11,365) = 17.1, *p* < 0.001
	group	*F* (123) = 4.10, *p* = 0.054	*F* (1365) = 30.5, *p* < 0.001
	age × group	*F* (11,253) = 1.00, *p* > 0.05	*F* (11,365) = 0.99, *p* > 0.05
Wire Hanging Test (Test 3)	age	*F* (11,253) = 11.0, *p* < 0.001	*F* (11,365) = 10.5, *p* < 0.001
	group	*F* (123) = 26.7, *p* < 0.001	*F* (1365) = 179.0, *p* < 0.001
	age × group	*F* (11,253) = 1.00, *p* < 0.05	*F* (11,365) = 2.16, *p* < 0.05

**Table 2 animals-10-00746-t002:** Results of statistical analysis in Test 4.

Foot Print Test (Test 4)		Repeated Measures Two-Way ANOVA	Two-Way ANOVA
		Alive Mice at 93 Weeks of Age	All Mice
Stride of forefoot	age	*F* (5120) = 16.4, *p* < 0.001	*F* (5182) = 18.0, *p* < 0.001
	group	*F* (124) = 16.4, *p* < 0.05	*F* (1182) = 4.45, *p* < 0.05
	age × group	*F* (5120) = 1.73, *p* > 0.05	*F* (5182) = 1.85, *p* > 0.05
Sway of forefoot	age	*F* (5120) = 10.8, *p* < 0.001	*F* (5182) = 12.2, *p* < 0.001
	group	*F* (124) = 4.15, *p* = 0.051	*F* (1182) = 3.86, *p* = 0.051
	age × group	*F* (5120) = 0.70, *p* > 0.05	*F* (5182) = 1.56, *p* > 0.05
Stance of forefoot	age	*F* (5120) = 2.48, *p* < 0.05	*F* (5182) = 3.29, *p* < 0.01
	group	*F* (124) = 0.245, *p* > 0.05	*F* (1182) = 0.0007, *p* > 0.05
	age × group	*F* (5120) = 1.30, *p* > 0.05	*F* (5182) = 1.16, *p* > 0.05
Stride of hindfoot	age	*F* (5120) = 8.51, *p* < 0.001	*F* (5182) = 9.06, *p* < 0.001
	group	*F* (124) = 14.4, *p* < 0.001	*F* (1182) = 8.26, *p* < 0.01
	age × group	*F* (5120) = 1.19, *p* > 0.05	*F* (5182) = 1.08, *p* > 0.05
Sway of hindfoot	age	*F* (5120) = 2.70, *p* < 0.05	*F* (5182) = 4.65, *p* < 0.001
	group	*F* (124) = 13.4, *p* < 0.01	*F* (1182) = 10.9, *p* < 0.01
	age × group	*F* (5120) = 2.01, *p* > 0.05	*F* (5182) = 1.31, *p* > 0.05
Stance of hindfoot	age	*F* (5120) = 4.56, *p* < 0.01	*F* (5182) = 3.39, *p* < 0.01
	group	*F* (124) = 0.85, *p* > 0.05	*F* (1182) = 1.59, *p* > 0.05
	age × group	*F* (5120) = 1.37, *p* > 0.05	*F* (5182) = 1.50, *p* > 0.05

**Table 3 animals-10-00746-t003:** Results of statistical analysis in Test 5.

Locomotor Activity (Test 5)		Repeated Measures Two-Way ANOVA	Two-Way ANOVA
		Alive Mice at 96 Weeks of Age	All Mice
Locomotor activity	age	*F*(8168) = 8.75, *p* < 0.001	*F* (8268) = 6.76, *p* < 0.001
	group	*F* (121) = 0.032, *p* > 0.05	*F* (1268) = 0.083, *p* > 0.05
	age × group	*F* (8168) =0.95, *p* > 0.05	*F* (8268) = 0.554, *p* > 0.05
Center time	age	*F* (8168) = 29.0, *p* < 0.001	*F* (8268) = 26.0, *p* < 0.001
	group	*F* (F(121) = 1.20, *p* > 0.05	*F* (1268) = 0.544, *p* > 0.05
	age × group	*F* (8168) =4.20, *p* < 0.01	*F* (8268) = 2.83, *p* < 0.001

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
