# Peer review of "Influence of Housing Systems on Physical, Emotional, and Cognitive Functions with Aging in DBA/2CrSlc Mice"

_animals, 2020, doi:10.3390/ani10040746_

Round 1

Reviewer 1 Report

The results of this study may be of use to PIs or directors of animal research facilities when confronting a major purchase of new cage systems for a given laboratory or animal care facility.  A major strength of this study is the wide array of physical, physiological, and behavioral measures taken of mice from the two housing systems.

My individual suggestions are listed below:

  1. The title should be changed to "Influence of housing systems on physical, emotional, and cognitive functions during aging in DBA/2CrSlc mice". The word "rearing" is too ambiguous.
  2. Housing should be substituted for rearing throughout the manuscript.
  3. Line 66:  species should be strain
  4. line 155 time staying in the white area

General comments:

  1. Dangling is actually more about physical activity and exercise that is available in one type of cage but not the other.  This should be clarified in the manuscript.
  2. It would have been helpful to quantify activity levels over a 24-hr period in mice housed in the two cage types.
  3. A justification should be provided for using DBA/2 mice.  Why this strain and not another strain?  In addition, why not include female mice as has been recommended by several funding agencies.

Reviewer 2 Report

Comparative studies between different caging systems are always of high interest. It is also well known that different rearing systems may create differences on physiology and behaviour of housed animals and may have an impact to the experimental results.

The novelty of this manuscript is that the authors are focusing on two different rearing systems the IVC and Chamber system. There is no doubt that the authors spend a lot of time and energy for this long term study. Unfortunately there are major problems which are linked to the poor design of the study and to my opinion influence the reliability and the repeatability of the results.

The first important issue is that the rearing conditions were changed during the study. From a group of six animals in three cages, the authors switched to 2 cages of group-housing and 11 or 12 isolated cages. So the question which is raised is: what kind of comparisons can be done as far as caging conditions are changed. This is also seriously linked with the question which is the experimental unit which is used (Ref. Festing MF, Altman DG  ILAR, 2002 (43)4:244-258).

The second equally important issue is the lack of environmental enrichment. From 6 to 14 weeks old the animals are reared without any enrichment device. This led to fighting problems and isolation of the animals. Only at the age of 19 weeks old environment was enriched. 

Of ethical concern is also the isolated caging of the animals for so long period.

Because of these the manuscript should be rejected

Reviewer 3 Report

The manuscript presents a comparison of two different laboratory caging systems on the physiology and behaviour of male mice. The study is very thorough and includes a large amount of physiological and especially behavioural parameters that were tested over a long time (about 1.5-2 years). As such, it provides a very thorough picture of how different laboratory holding systems (even if the difference can be perceived as relatively small by humans) can affect typical laboratory rodents throughout a large part of their lives, which is also typically the time that they are used in experiments. This knowledge is of utmost importance for a better understanding of how the laboratory environment may affect experimental results. The authors also further pointed out that some results obtained in different laboratory cage settings may not be comparable and that researchers have to be careful when interpreting such results.

Although I must commend the authors on the effort they put into data collection, there are some very large problems with the manuscript. First, it is very difficult to follow the aims and objectives of the study and it is often unclear why some things were measured. From the beginning, the authors should carefully consider what it is that they want to measure and why. At the moment, it seems to be just a list of things that were done because they could and the rationale for each of the measurements is lacking. This is further problematic as the results are not well discussed and the discussion appears to be primarily a repetition of the results. Further, the objective of aging appears to be an afterthought in the introduction, but yet it is one of the main aspects of the manuscript. However, I think calling it aging is not quite appropriate and the authors should think of another term or phrase. The authors used mice during the main part of their lives, whereas aging would imply a study looking at senescence. Please carefully consider the objectives and justifications in measuring specific things and include some general predictions, which can then be used in the discussion. Also, distinguish better between the different types of experiments and tests.

Second, I have major concerns with the statistical methods used in the present study. ANOVAs and t-tests are not sufficient for this data as the animals were measured repeatedly and as such data points in the different weeks are not independent from one another.  The authors should use repeated measures ANOVAs or mixed models. From the description of the statistics, it is not clear what was compared in the different tests and additional explanations should be added. Further, I do not think that all these variables covered the assumptions of the tests used (i.e. normality and homogeneity). In addition, the authors should consider using a PCA to combined variables measured during a specific behavioural test or a group of tests (e.g. the tests looking at climbing ability) to combine these into one or two components, which can then be compared between the ages, caging systems, etc.  This would also tighten the results and discussion and make it easier to see which variables are truly important when considering the different behaviours – it is currently very difficult to ascertain this, but it is very important for interpreting results in the present and other studies.

More specific comments:

Simple summary:  This needs to be rewritten. It feels very disjunct and repetitive and it is unclear what was measured and what the results are (too general and very little results). The conclusion is too general. How could the findings affect experiments?

Introduction:

In general, the authors rely a lot on studies from their own research group and other literature is missing. Further, much of the information is given without a context and it is unclear how it is important for the present study.

L.70-76: It is unclear why this is important.

  1. 88: It is unclear why and how aging is important in this study. For example, over what period were the mice tested.

Materials and Methods

In general, the information on each of the tests conducted is not enough. It is very difficult to understand how many of the tests were conducted without reading an additional 10 or more papers. This is simply too much and makes it almost impossible to evaluate the usefulness of each of the test and experiments. Some general information is required for each of the tests besides a reference,  which can then be included to give more specific details. The MM could be shortened in other places such as the times when measurements were done as they are in the supplementary table.

  1. 108-110: It is not quite clear what was happening with the mice here. Some were isolated, but then there were still fights. I am assuming that the mice were not put together again. Please be more specific when explaining why and when mice were isolated, how many were isolated and how many remained in groups. Further, group vs. single housing needs to be statistically evaluated and an effect of the housing regimes should be excluded.
  2. 120: How was food consumption measured?

L.124: This should not be part of the MM and should rather be placed in the introduction or discussion.

E.g. 133-136: Why were the tests conducted at such different times? Could they have affected each other? Tests should have been randomized and it should be discussed if and how the experimental set-up may have affected results.

  1. 179: Refer to the supplementary material.
  2. 161…: At this point, it is unclear what the purpose of these tests is, what the mice are supposed to do and what was measured. Please include a general description for each of the tests as references are not sufficient.

Results

Please include the full statistics for each of the tests conducted and not just p-values.

Figure 2: The y-axis should read food intake.

  1. 196: Include a measure of variance. These two sentences appear to be contrary to each other and it is not clear what the results are.
  2. 215: Here and at other places in the results, please indicate what the differences were – i.e. in which group and at what age was it higher.
  3. 233: The purpose of this test is not clear. What is the relation to the climbing test? It is also very difficult to see that there are really differences and other, more appropriate tests may in fact not find any significant differences.
  4. 234-237: Results should only include the actual results and no descriptions of tests or evaluation of results. This should be in the MM.
  5. 263: The sentence structure is not correct. Please rewrite.
  6. 317: Is this really a recognition test? Results and general understanding provided in the manuscript would indicate that it is rather a novel-object test.
  7. 326: The entire section is missing the statistics. What tests were used?
  8. 333: Why would you expect a difference between groups? Both were fed and housed (bedding) the same. The discussion seems to give a hint but appears not specific enough.
  9. 337: What are “futures”?
  10. 344: Replace remarkable with pronounced.
  11. 349: These are percentages and not ratios.

Discussion

The discussion is primarily a repetition of the results and is lacking a lot of relevant literature. Results should be put into the context of the aims and objectives of the study and results should be explained in the context of predictions and other studies. Authors need to more thoroughly investigate what these results mean for other studies using the literature.

  1. 370: The data should be included at least as supplementary material.
  2. 414, 428 and others: What was the rearing system and why was it different between the groups? Please include this in the MM as it seems to have a considerable effect on the results. The discussion of this is too vague.

Round 2

Reviewer 2 Report

Thank you for the provided clarifications. The quality of the manuscript was improved.

A suggestion in relation to figure 1: A better photo of Chamber system should be used

Author Response

Reviewer'2 comment: A suggestion in relation to figure 1: A better photo of Chamber system should be used

Response: Thank you very much for your suggestion. I have replaced the pictures of housing systems in Figure 1.

Reviewer 3 Report

The changes the authors have made to the manuscript “Influence of housing systems on physical, emotional, 2 and cognitive functions with aging in DBA/2CrSlc 3 mice” have improved the readability of the manuscript but I do not think that they have been far-reaching enough. Additions to the Materials and Methods have improved the understanding of the tests conducted and the aims and objectives of the different tests are clearer. Nevertheless, this could be further improved although the clarity is probably lacking due the authors struggling with the English language. I would, therefore, highly recommend that the authors ask somebody to help them with that. I will also point out a few instances where improvements need to be made (not all-inconclusive) in my later comments. I still have major concerns in regards to the statistics used: The statistics employed are not suitable for the analysis of differences between age classes. I understand the problem the authors face in regards to animals dying; however, the data points are simply not independent and as such cannot be evaluated using the statistical methods in the manuscript. This is especially important when looking at behavioural traits as the authors have done primarily in the manuscript. I would, therefore, recommend that the authors use only the animals that survived throughout the experiments when evaluating effects of age on any of the variables tested and use repeated measures ANOVA or mixed modelling approaches for these types of analyses. Further, the authors do not indicate if the variables are normally distributed and homoscedastic, etc. I do not believe that all the different variables are indeed following the assumptions of ANOVAs (or t-tests) and as such, those tests are likely not suitable for all of the comparisons done in this study. Other comments: L. 16: The sentence structure is strange especially in regards to the breeding environment of humans. L. 20: Please rewrite to make the meaning clearer. L.27-28: This is very general and the meaning is not quite clear. L. 64-67: The sentence is very long and confusing and should be rewritten. L. 69: Please include example studies and references to support this. L. 74: Change to “one of the most commonly used tests” L. 75: Change to “a black and a white box” L. 77-78: Change to “spends in the white or the black box”, remove “in the equipment” L. 84-87: The sentence is very long and should be split. L. 106-108: The sentence is too long and difficult to follow. L. 136: What behaviours were evaluated? As some behaviours are not possible for single housed animals (e.g. allogrooming, huddling), some of the behaviours were likely different between the single and group housed animals. L. 138: How often was the dishcloth gourd renewed/replace? L. 162-180: The explanations for test 1 and test 3 could probably be combined to save space as the general test structure and measurements are very similar. It is unclear why the measurements done for test 1 to 3 were so different (measurements in second vs. a scoring system) considering the similarities between the tests. L. 215: The habituation part is very similar to the tests 6 and 7 and it is likely that the different tests would affect each other especially considering the effect of age. It is likely that the animals would become more and more habituated to these types of tests especially as they are being conducted so frequently possibly affecting their behaviour over time and with increasing age. The authors should take that into account when discussing the results (reduction of anxiety like behaviours with age). L. 227-235: The description of what was measured here is very confusing. L. 271-273: Were these body mass changes significant? How could the fighting have influenced the results besides for body mass (behavioural measures) shown in the study (Discussion)? L. 280 and throughout the results: Aims of the different tests should not be included in the results. These should be made clear in the Materials and Methods and possibly in the introduction. Please remove these throughout the results section and focus solely on the actual results (descriptive and statistical tests/comparisons). L. 286 and throughout the results: Please give the entire test statistics and not just p-values (except for posthoc results). L. 287: You could give overall means and errors in the text. Figure legends (overall): I commend the effort of the authors in giving all the sample sizes; however, it is rather confusing and makes the legends unnecessarily long. If samples sizes (per age class and for each of the housing conditions) need to be given, it may be better to show those in a table or include them directly in the figures. Age comparisons should only be done using the animals that survived throughout the entire experimental time though (see above). L. 298 and throughout: Please indicate what the actual observed differences were. Was a variable higher or lower in one of the housing conditions compared to the other? Was there an increase or a decrease in a variable with age? The figures will support these things said in the results section but just saying that there is a significant difference is not enough (the results section is extremely boring to read as it is not very informative). Figure 5: Please indicate on the y-axis what s is. I assume it should be Time in seconds (Time (s)). Y-axis for Figure 5b should be “Degree of climbing potential”. Some of the asterisks in Figure 5c are difficult to see. L. 327-328: Should be part of the MM and not the Results. L. 329-334: The entire paragraph should be shortened. Some of the results could be easily combined, e.g., there is never a significant interaction. L. 370-371: This sentence is unnecessary as a reference to the figures can be easily included in the subsequent sentences as done, for example, in line 378. Similarly, the sentence in lines 389-390 can be removed. L. 419-420: Should be part of the materials and methods (or discussion) and not the results. L. 441: Something in regards to the microbiome changes with age in the direction of the arrow, it is, however, unclear what this change is. How did the microbiome change with age? L. 472: Why would walking/jumping be different in the two housing systems? Mice in both of the systems were enriched, or? Only the ability to climb (on the lid) should be different, or? L. 483: Remove “and decreased”. L. 498: Add at the end of the sentence “associated with lower exercise”. L. 503: Add “better at climbing tasks than”. L. 507: I think it should be stressed that it is not just daily exercise (which could also be in the form of e.g. wheel running) but the type of exercise that may play a role (dangling/climbing on lid). L. 524: Why would especially younger but not older mice be affected? L. 542-544: Could the changes in anxiety behaviours with age be due to the repeated testing? Please include this in the discussion. L. 549: Why would there be a difference in anxiety between the housing systems? This is not clear from the discussion. L. 582: Please include studies in rodents in addition to the human studies to allow for better comparisons. L. 580-582: It is unclear what dangling has to do with a mouse approaching an unfamiliar object. This should be further explained. L. 603-606: This has nothing to do with the microbiome that is mentioned just preceding it. This part would probably be better placed when discussing the behaviours.
